# Extraordinary Room-Temperature Tensile Ductility of Pure Magnesium

**DOI:** 10.3390/ma12233813

**Published:** 2019-11-20

**Authors:** Xinghao Du, Haitao Chang, Cai Chen, Xiaofeng Huo, Wanpeng Li, Jacob C. Huang, Guosheng Duan, Baolin Wu

**Affiliations:** 1School of Materials Science & Engineering, Shenyang Aerospace University, Shenyang 110136, China; haitaochang652@163.com (H.C.); chencai01@siju.edu.cn (C.C.); huoxf1995@163.com (X.H.); wubaolin@sau.edu.cn (B.W.); 2School of Materials Science & Engineering, Shanghai Jiao Tong University, Shanghai 200240, China; 3Department of Materials Science & Engineering, City University of Hong Kong, Kowloon 999077, Hong Kong, China; wanpengli2-c@my.cityu.edu.hk

**Keywords:** pure Mg, multiple-axial forging, mechanical properties, strain hardening, pyramidal dislocations

## Abstract

Room-temperature tensile behavior and associated deformation mechanisms of multiple-axial forged (MAFed) pure Mg has been investigated. The as-MAFed Mg, with a coarsely recrystallized structure, exhibited a balanced strain-hardening behavior with strain, resulting in extraordinary mechanical properties with high ultimate stress (~200 MPa) and extensive true strain (~0.30). The observation on the microstructural evolution suggests that the balanced strain-hardening behavior is correlated with de-twinning behavior cooperated with pyramidal <c + a> dislocations at the plastic straining stage.

## 1. Introduction

It is known that magnesium with a hexagonal close-packed (HCP) crystal structure, as well as a c/a ratio slightly less than the ideal value of 1.633, usually exhibits poor room-temperature formability for making structural components through plastic processing [1,2]. The insufficient plasticity of Mg is attributed to the dominant basal (0001) <11$\bar{2}$0> slip owing to its much lower critical resolved shear stress (CRSS) compared with dislocations slip on other crystallographic planes [3,4]. In addition, texture usually exhibits an unfavorable effect on formability. Owing to this, texture control is always valued in processing [5]. In order to improve the room temperature plastic deformation in Mg alloys, it is necessary to activate the pyramidal dislocations <c + a> slip and/or deformation twining [6,7]. In particular, there are five independent pyramidal slip systems for <c + a> dislocations, therefore, if the slip of <c + a> dislocations were available, it should significantly influence the mechanical response of the HCP-type Mg and its alloys [8,9].

In an attempt to activate the slip of <c + a> dislocations of Mg and Mg alloys at room temperature, several efforts have been made so far. One way is to add rare earth elements (RE) to form Mg-RE alloys; the beneficial function of RE on the activation of <c + a> dislocations is realized by shortening the c/a ratio [10] and changing the atomic binding states [11,12] as well as decreasing sessile I1 stacking faults energy [13,14]. For pure Mg, the improved tensile ductility with a strain of up to 0.27 can be achieved in the refined microstructure of pure Mg experienced by the equal channel angular pressing (ECAP) process [15]. In addition to this, the <c + a> dislocations or contraction twins can be activated only under the high-strain-rate compression process, when the crystallographic c-axial is parallel to the stress direction [16,17]. Therefore, there has been great progress made in improving the poor plasticity of Mg alloys, but how to realize the room-temperature remarkable tensile plasticity of pure Mg through the activation of <c + a> dislocations is still a challenging topic.

Recently, pre-twinning was reported and summarized as an effective method to enhance the performance of magnesium alloys by Song et al. [18]. It indicates that pre-twinning can play an important role in improving strength, formability, and fatigue, as well as texture and dislocation slips control. It was found that pre-twinning, in which (10–12) extension twins are induced by pre-rolling along the transverse direction (TD) of the rolled plates, can effectively reduce the average r-value [19] and rolling capacity [20] due to the reorientation of the twins that is favorable for the activation of multiple deformation modes. Among the multiple deformation modes, pyramidal <c + a> slip is the most important because of its strain accommodation role along the c-axis direction. An enhanced plastic strain has been observed in pure Mg during the high-strain-rate compression process at room temperature, during which extensive pyramidal <c + a> slips have happened in the deformation twinning regions [21]. It was presented that the interaction of activated pyramidal <c + a> slips and twins caused obvious strain hardening behavior, contributing to the enhancement of compressive plasticity. This inspired us to probe the room-temperature tensile deformation behavior of pure Mg with an initial high-density twinning microstructure. To do this, in this study, we prepared pure Mg bulk samples with high-density twins by using multiple-axial forging (MAF). As expected, the as-multiple-axial forged (MAFed) pure Mg samples exhibited promising mechanical response tensioned at room temperature under low strain rates. 

## 2. Materials and Methods

An Mg ingot of commercial purity (99.9%) was used in this study. The ingot was machined for MAF into samples with a size of 50 × 50 × 50 mm. First, the cubic samples were heated up to 693 K in an electric resistance furnace for 2 h. Then the MAF process was conducted using an industrial air pneumatic hammer machine with a load gravity of 750 kg. During MAF, the forging direction was changed by 90° pass-by-pass, as described in the literature [22,23]. A small pass strain of Δε = 0.05 was employed in each pass. After 9 passes had been carried out on the samples, they were re-annealed at 693 K for 0.5 h. The above forging procedure was repeated four times to achieve a maximum cumulative strain of ∑Δε = 1.80. Finally, the samples were cooled down in air and denoted as “as-MAFed”. Tensile specimens with a dog bone shape (10 mm gauge length and 2 × 3 mm^2^ cross section) were machined from the central part of the MAFed samples along the transverse direction (TD) and longitudinal direction (LD). The tension tests were carried out on a Sans type tensile testing machine with an initial strain rate of 1 × 10^−3^ s^−1^ at room temperature.

Electron backscatter diffraction (EBSD) and X-ray diffraction (XRD) were used for microstructure observation and texture measurement. EBSD was performed using a JEOL-JSM-7001F (JEOL, Tokyo, Japan) field emission scanning electron microscope equipped with an automatic orientation acquisition system (Oxford Instruments-HKL Channel 5). The specimens for EBSD were mechanically polished followed by an electrochemical polish in commercial AC_2_ solution. Macro-texture was measured by XRD (Bruker D8 Discover instrument, Beerlika, MA, USA) using the Schultz reflection method. For the detailed microstructure observation, a JEM-2000EX (JEOL, Tokyo, Japan) transmission electron microscope (TEM) was used. The foils for TEM observation were thinned at 10 V and −40 °C by the twin-jet polishing technique using an electrolyte consisting of 1% HClO_4_ and 99% methanol.

## 3. Results

### 3.1. Characterization of Macrostructure after MAF

Figure 1a,b presents the microstructure with different magnifications, and Figure 1c represents the (0002) and (10$\bar{1}$0) incomplete pole figures of the as-MAFed Mg samples. It can be seen that the size distribution of the recrystallized microstructure was quite uneven, and some recrystallized grains were very coarse with a mean grain size of ~50 μm, owing to the fact that pure magnesium with a single-phase microstructure could easily grow up during the recrystallization process. Furthermore, most of the grains were packed with high-density twins. It is discerned that the high strain rate (20 s^−1^) of the forging process can accelerate the formation of twins [17]. Figure 1c shows that the as-MAFed Mg exhibited a relatively random texture, and the maximum intensity of 5.34 was weaker than the texture intensity of 7.9 in room-temperature MAFed AZ61 alloys [16]. The formation of the weakened random texture in this work is similar to other research [23,24].

### 3.2. Tensile Behavior of as-MAFed Mg Samples at Room Temperature

The engineering stress-engineering strain of as-MAFed Mg and true stress vs. the true strain curves of the as-MAFed as-extruded [25], as-rolled [26], and as-cast Mg [27] are presented in Figure 2a,b, respectively. It is interesting to note that the as-MAFed samples, both in LD and TD, exhibited a very similar mechanical response, indicating that the MAFed Mg has weak anisotropy in nature. The true stress-true strain curve in Figure 2b shows that the as-MAFed Mg exhibits a remarkable combination of tensile stress (150 MPa) and tensile strain (~32%), which is superior to the results of previous studies [25,26,27]. Following the perfectly elastic deformation stage, the sample entered the plastic deformation stage, with obvious strain hardening. The flow stress was seen to increase to about 200 MPa at a true strain of 0.30; a very promising combination of strength and plasticity for pure Mg tension-loaded at low strain rates and room temperature. By comparison, the as-extruded, as-rolled, and as-cast Mg exhibited inferior mechanical response, with much lower tensile strains. The work-hardening rate, defined as the rate of change of true stress as a function of true strain, Θ=dσ/dε, is also presented in Figure 2b. It is observed that the as-MAFed Mg exhibited soundly balanced strain-hardening behavior with strain. On the contrary, the as-extruded, as-rolled, and as-cast Mg exhibited a continuously decreasing strain-hardening rate. Presumably, the gain of remarkable strength and plasticity of MAFed Mg is associated with the beneficially balanced hardening behavior.

### 3.3. Characterization of Microstructural Evolution during Tension Process by EBSD 

EBSD was conducted to monitor the structural evolution upon tensile loading, as presented in Figure 3. Figure 3a is the band contrast map of as-MAFed Mg before loading, in which the boundaries were identified with <11–20> 86.4° (red lines), <10–10> 60° (yellow lines), <11–20> 7.4° (light blue lines), and <8–1–70> 60.4° (pink lines) misorientations, respectively, with the aid of Channel 5 software. According to the misorientations data given in the literature [5], the red line represents the boundaries between tension twins and parent grains. The other three kinds of line represent the boundaries between different tension twin variants. Figure 3b,c shows the corresponding misorientation angle distribution profile and scattered {0001} pole figure. From the misorientation angle distribution profile (Figure 3b), it can be seen that the misorientation angles were respectively distributed below 5°, which corresponds to crystallites with low angle grain boundaries (LAGBs) and variants boundaries with a misorientation of <11–20> 7.4°. Misorientation angle peak around 86.4° corresponds to tension twins in Figure 3a. In addition, a lower misorientation angle peak around 60° corresponds to the boundaries between different pairs of twin variants with misorientations of <10–10> 60° (yellow line) and<8–1–70> 60.4° (pink line), respectively. No compression twins can be found in the as-MAFed Mg.

Figure 3d–f shows, respectively, the band contrast map, misorientation angle distribution profile, and the scattered {0001} pole figure of the MAFed sample after a tensile strain of 0.20, which corresponds to the balanced work-hardening stage. Compared with Figure 3a, it is noted in Figure 3d that the amount of tension twins decreased dramatically, indicating that intensive de-twinning behavior occurred, accompanying the plastic deformation. In addition, some <11–20> 56° ((10–11) <10-1-2>) compression twins (blue lines) and <11–20> 38° ({10–11}–{10–12}) double twins (light green) formed. Due to the de-twinning, the boundaries between the different variants decrease dramatically. Furthermore, from the (0002) pole figures of Figure 3c,f, it is discovered that, after the straining of 0.20, the crystallographic orientation has been changed from a random to a moderately sharp state. The result coincides with previous research [28], which states that texture would be formed because the amount of grain rotation increases rapidly when the strain reaches 0.1. It is noted that similar de-twinning behavior of the as-MAFed AZ31 alloy during room-temperature deformation has also been reported by Sarker et al. [24]. They argued that the de-twinning behavior was associated with the intensive interaction between the multiple slip of dislocations and twins. The de-twinning behavior is beneficial for the alloy to achieve remarkable elongations by enhancing the strain hardening rate [24].

In this work, firstly, initial random texture has formed during the multiple-axial forging process, as shown in Figure 1c and Figure 3c. From the EBSD measurement results (as shown in Figure 4), the Schmid factors (SFs) for the basal <a> slip system {0001} <11$\bar{2}$0>, the pyramidal <c + a> slip system {10$\bar{1}$1} <11$\bar{2}$0>, and the pyramidal <c + a> slip system {11$\bar{2}$2} <11$\bar{2}$3> were estimated to be 0.36, 0.38, and 0.39, respectively. This means that the random texture of the MAFed Mg makes the SF very high for all slip systems. Hence, owing to the lowest CRSS for the basal <a> slip [1,2], it should be activated preferentially and plays an important role in accommodating the strain in the initial plastic deformation stage. This corresponds well to the mechanical response in the initial stage of deformation, in which a very low yield stress (25 MPa) was obtained. Koike et al. [29] also showed that in large-grain magnesium, basal slip facilitates microscopic yielding at very low stress. In order to accommodate further deformation, other slip systems should get activated.

### 3.4. Characterization of Microstructural Evolution during Tension Process by TEM

Next, TEM weak-beam dark-field observation was used to observe the details in the microstructure of the as-MAFed samples strained at the uniform plastic deformation stage. Figure 5a,b presents the images taken at g = (0002) and g = (11$\bar{2}$0) of an area in the interior of a grain in the specimen with a true strain of 0.20. It can be clearly seen from Figure 5a,b, that the dislocations slips were considerably active, and multiple slips could be observed in the deformed regions. No twins were observed. In particular, several dislocations (marked by pink arrows in Figure 5a,b are visible in the two imaging conditions. According to the  g·b=0  invisibility criterion, these dislocations must be of the <c + a> type. This shows that pyramidal <c + a> slips have been activated during the room-temperature tensile process. Furthermore, a region composed of a colony of sub-grains could be observed in the specimen, as marked by the yellow words in Figure 5c. The sub-grains were surrounded by high-density dislocations regions. The dissociated <a> segments from the <c + a> dislocations tend to cross-slip, thus promoting the formations of sub-grains [16]. This provides further evidence of the activation of <c + a> dislocations.

The activation of <c + a> dislocations have been observed when the texture of the Mg alloys is such that most grains are favorably oriented for *c*-axial compression [15,16,17]. Obviously, this is not applicable in this work, in which tension stress is employed. Here, the observed multiple slips, including the <c + a> dislocations, can be understood if the equivalent resolved shear stress, which is determined by CRSS, SFs (orientation), and applied stress, could reach similar levels for all dislocations slips. On one hand, in this study, the coarse grain (~50 μm) polycrystalline Mg might possess a non-basal CRSS about one-order of magnitude bigger that for basal slip, according to the calculation on the basis of the Schmid factor with respect to the loading direction [29,30,31]. Furthermore, Huchinson and Barnett [32] have shown that if the soft annealing polycrystalline Mg materials were deformed to a moderate strain, the CRSS_non-basal_/CRSS_basal_ ratio would decrease further to ~2. In this work, due to the easy slip of basal dislocations from the initial random texture as well as the hindering effects of pre-existed profuse twins, the dislocations density would increase rapidly with the increase of plastic strain (as shown in Figure 5), and this will definitely reduce the CRSS_non-basal_/CRSS_basal_ ratio to an expected scale. On the other hand, from the EBSD measurement results of the samples deformed up to a strain of 0.20 (as shown in Figure 6), the SFs for the basal <a> slip system {0001} <11$\bar{2}$0>, the pyramidal <c + a> slip system {10$\bar{1}$1} <11$\bar{2}$0>, and the pyramidal <c + a> slip system {11$\bar{2}$2} <11$\bar{2}$3> were estimated to be 0.34, 0.373, and 0.419, respectively. Hence, it is reasonable to conclude that the multiple slips composing of non-basal <c + a> dislocations have been activated during the plastic deformation process.

## 4. Discussion

Based on the above observations, here we can discuss the underlying mechanism responsible for the enhanced mechanical properties of the as-MAFed Mg tensioned at room temperature. It is common to accept that the plasticity of basal textured-Mg is attributed to the dominant basal (0001) <11$\bar{2}$0> slip owing to its much lower critical resolved shear stress (CRSS) [33]. It is known that this kind of dislocations-slip-dominated plastic deformation results in continuously decreasing strain-hardening rates [34]. In such a case, the pre-mature failure is caused due to its inability to avoid the plastic instability in the early plastic stage. This has happened in the wrought Mg samples when tension testes were carried out with the tensile direction perpendicular to the <c> axial of most grains [25,26]. Previous studies have, as has this work, proved that, to obtain a good balance of strength and ductility of metals, balanced strain-hardening behavior is necessary [35,36,37]. Furthermore, several methods have been explored to strengthen and toughen the Mg base alloys simultaneously [38,39]. It is of great interest to observe that pyramidal <c + a> dislocations have been activated due to the synthesized effects of initial coarsely grained structure with randomized texture as well as the pre-existed profuse twins of MAFed pure Mg during the tension process. Most importantly, it brings about a rapid work hardening. The rapid work hardening originates from two facts: One is that the <c + a> dislocations tends to dissociate to <a> and <c> segments, and the work hardening can be attributed to the formation of the immobile <c> dislocation segments, which strongly prevents other mobile dislocations from passing through, i.e., the formation of the forest interactions between the non-basal dislocations [40,41,42,43]; the other is the intensive interaction of non-coplanar dislocations and pre-existed mechanical twins, as suggested in paper [24]. This process would result in de-twining behavior. Accordingly, the softening mechanism from dislocations slip can be counteracted by the strengthening mechanism from both the de-twinning behavior as well as the activation of pyramidal <c + a> dislocations, resulting in a good balance of strength and ductility.

## 5. Conclusions

In conclusion, a good balance of strength and ductility in commercially pure Mg with a HCP crystallographic structure has been achieved successfully by using the multiple-axial forging method. The good balance of strength and ductility of MAFed Mg is associated with the balanced strain-hardening behavior, which is stemmed from the activation of de-twinning and non-basal pyramidal <c + a> dislocations slip. The introduction of high-density mechanical twins during the multiple-axial forging process plays a significant role to provide strong strain hardening in the plastic deformation stage.

## Figures and Tables

**Figure 1 materials-12-03813-f001:**
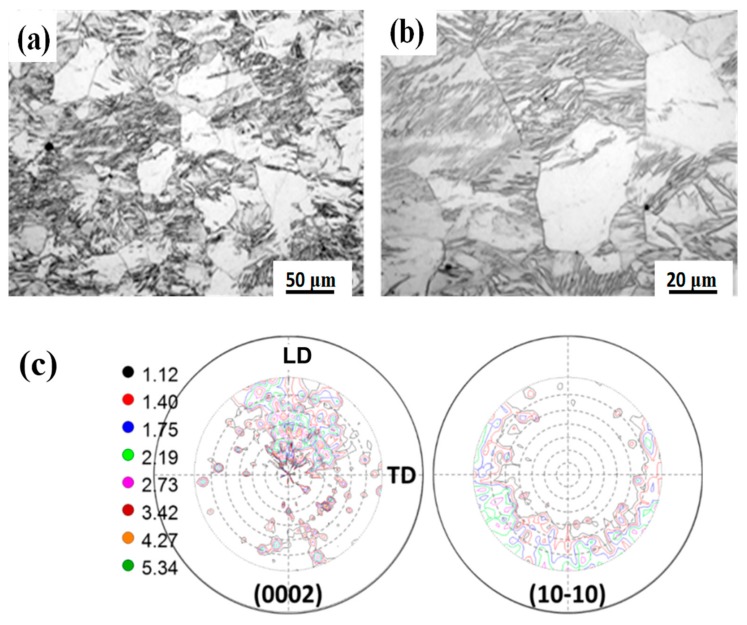
Showing (**a**,**b**) the microstructure of the multiple-axial forged (MAFed) Mg with different magnifications, and (**c**) the (0002) and (10$\bar{1}$0) pole figures of the MAFed samples, in which LD represents longitudinal direction; TD represents transverse direction.

**Figure 2 materials-12-03813-f002:**
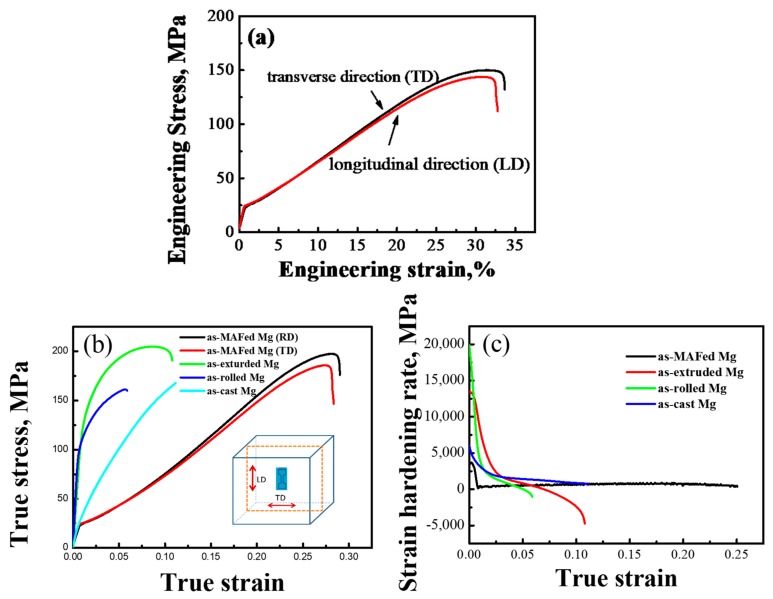
(**a**,**b**) Showing the engineering stress-engineering strain and true stress-true strain curves of as-MAFed, as-extruded, as-rolled, and as-cast pure Mg at a strain rate of 1 × 10^–3^ s^–1^ at room temperature; (**c**) shows the true strain hardening rate as a function of true strain of the as-MAFed, as-extruded, as-rolled, and as-cast pure Mg.

**Figure 3 materials-12-03813-f003:**
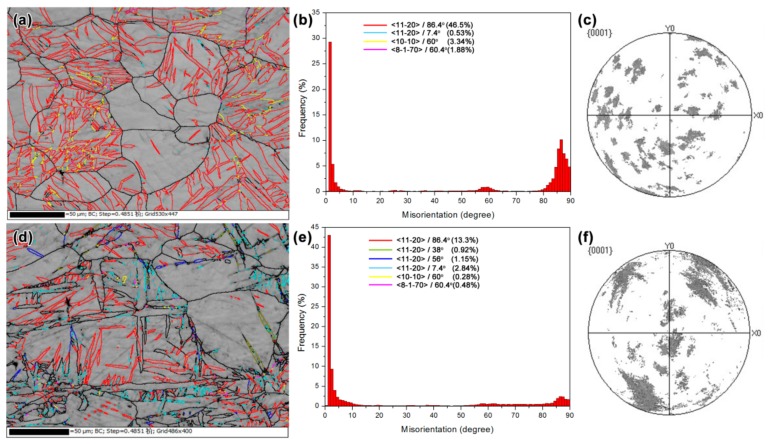
Showing the electron backscatter diffraction (EBSD) band contrast maps of the distribution of twin and misorientation angle as well as the (0002) pole figures of the as-MAFed sample (**a**–**c**) and the deformed sample with a true tensile strain of 0.20 (**d**–**f**).

**Figure 4 materials-12-03813-f004:**
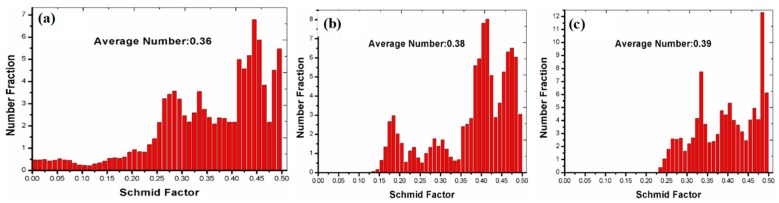
The magnitude of Schmid factors of the various sliding systems for the as-MAFed pure Mg: (**a**) The basal <a> sliding system ({0001} <11–20>; (**b**) the pyramidal <c + a> sliding system {10–11} <11–20>; and (**c**) the pyramidal <c + a> sliding system {11–22} <11–23>.

**Figure 5 materials-12-03813-f005:**
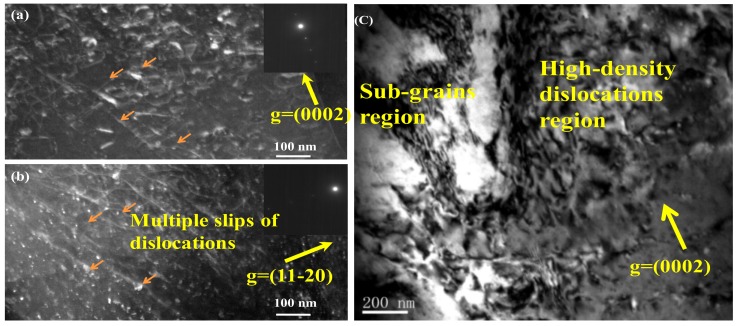
TEM weak-beam images taken from specimens deformed up to a strain of 0.20; (**a**,**b**) show the activated dislocations type, and (**c**) shows the microstructural morphology.

**Figure 6 materials-12-03813-f006:**
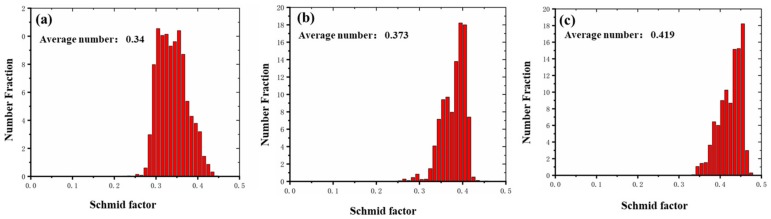
The magnitude of the Schmid factors of the various sliding systems for the as-MAFed pure Mg, deformed up to a strain of 0.20: (**a**) The basal <a> sliding system ({0001} <11–20>; (**b**) the pyramidal <c + a> sliding system {10–11} <11–20>; and (**c**) the pyramidal <c + a> sliding system {11–22} <11–23>.

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
