# Peer review of "Extraordinary Room-Temperature Tensile Ductility of Pure Magnesium"

_materials, 2019, doi:10.3390/ma12233813_

Round 1

Reviewer 1 Report

This manuscript describes the extraordinary room-temperature tensile ductility of pure magnesium. The manuscript is well organized. However, the claims of this manuscript are not fully supported by the key experimental data. My comments, concerns, and questions are as follows.

1.Please show nominal stress-strain curves of Figure 2 (a) as well. It is better to show the nominal stress-strain curves for understanding the tensile strength and uniform elongation. Are the results obtained in this study certainly “extraordinary” compared with the results in previous studies, for example, C.H. C´aceres, A.H. Blake / Materials Science and Engineering A 462 (2007) 193–196 ?

2. Lines 129 and 166: Please show the data as evidence since these are key results in this study.

3. Line 70: -40oC ==> -40 oC.

4. Lines 102 and 104: MATed ==> MAFed.

5. Line 126: (002) ==> (0002).

Reviewer 2 Report

The authors present results for quasi-static deformation and materials characterization of commercially pure Mg that has been multi-axially forged (MAFd) to finite extent. The authors perform EBSD and TEM to quantify the twin and dislocation evolution as a result of the processing and deformation. The topic of improving the response of Mg by pre-twinning is not a new area (see comprehensive review 5 years ago by Song et. al., M&D, 2014), so publication of results must meet some standard for presentation of new results or new analysis of existing results. Unfortunately, in the opinion of this reviewer, this work does not meet this level. Although I recommend rejection of the paper, if the authors are to look into the area further the suggested issues should be addressed.

Issues to address

In Figure 1 the authors compare the response of the material when it is loaded along a direction that twins versus directions that do not twin. Mg is known to have poor ductility along non-twining directions. The result of this work is that an enhanced ductility and UTS is obtained by pre-twinning, but if the authors compare to loading along a direction where the material twins significant gains will not be realized.

The authors seem to have neglected or have not been familiar with a large body of work in the pre-twinning literature spanning from 2012-2019. Based on the selected references the work appears to be new, but in light of other material the conclusions are not surprising or novel.

The authors assert that there is a combination of tension and compression twins based on misorientation angles. Although there are not other twin variants near 86 degrees, the {1012}-{1012} double twins occur at 60 degrees and can be confused with compression twins. The authors need to use axis-angle description (see Nave and Barnett, Scripta Mat, 2004 on how to do this) to distinguish whether the 56 degree misorientation corresponds to compression or double tension twins. Adjust conclusions accordingly. 

References to look at

Song, Bo, et al. "Improvement of formability and mechanical properties of magnesium alloys via pre-twinning: A review." Materials & Design (1980-2015) 62 (2014): 352-360.

Nave, Mark Denis, and Matthew Robert Barnett. "Microstructures and textures of pure magnesium deformed in plane-strain compression." Scripta Materialia 51.9 (2004): 881-885.

Round 2

Reviewer 1 Report

Figure captions of Figure 4 and Figure 6 are same.

Please state the difference of these two figures clearly in the captions.

Typografical errors can be still seen in the manuscript.

For example,

In Fig. 1 (c): LFD => LD

Line 171 (c,d,f) => (d,e,f)

Please carefully check the manuscript again.

Author Response

Dear reviewer:

Thanks for the suggestion. We have corrected the manuscript according to this suggestion one by one. Please see the attched file. 

Reviewer 2 Report

The authors did address most of the concerns. In figure 2b-c it is still confusing that the authors compared their mechanical behavior with magnesium alloys oriented so that they do not twin. It would be beneficial for the reader to compare your results with the mechanical response of mg alloys that are expected to twin.

Author Response

Dear reviewer:

Thanks for the suggestion. We have modified the paper according to your suggestion. Please see the attched file.

Thanks.

Round 3

Reviewer 2 Report

The authors have still not included ref 27 stress-strain response in Figure 2 b/c, which will help the reader understand that indeed the authors have improved the ductility but not affected the ultimate tensile strength. This is something that should be added, but I will leave this up to the reviewers.

The authors have still decided not to analyze whether or not compression twins exist in the sample. They state "From the misorientation angle distribution profile (Fig.3(b)), it is seen that the misorientation angles were mainly distributed below 5° corresponding to crystallites with low angle grain boundaries (LAGBs), some distributed at 56°, indicating that a little compression twins with misorientation of <11-20> 56 degrees exists in the microstructure, which we did not identify."

The above statement is not true unless the authors actually show that this is the case. Almost all EBSD software is able to determine the axis angle orientations. Other post-processing software is able to analyze this. The reviewer has pointed out how to do this. See Table 2 in Nave, Mark Denis, and Matthew Robert Barnett. "Microstructures and textures of pure magnesium deformed in plane-strain compression." Scripta Materialia 51.9 (2004): 881-885.

Author Response

Dear reviewer:

Thanks for the valuable suggestion. We have make a feedback one by one. Please see the attached file.

with best

Round 4

Reviewer 2 Report

n/a